# Dynamic Comprehensive Benefit Evaluation of the Transnational Power Grid Interconnection Project Based on Combination Weighting and TOPSIS Grey Projection Method

**Jinying Li, Jiaming Xu * and Xin Tan**

Department of Economics and Management, North China Electric Power University, Baoding 071003, China;
51850875@ncepu.edu.cn (J.L.); 2162218023@ncepu.edu.cn (X.T.)

**\*** Correspondence: 2172218050@ncepu.edu.cn; Tel.: +86-10-752-5125

**Abstract:** With the rapid development of the global economy, the interconnection of power grids has become an objective law and a trend of the power industry development. The implementation of power grid interconnection projects, especially transnational power grid interconnection projects, will bring us substantial benefits. To demonstrate these benefits comprehensively, we designed a comprehensive evaluation index system with multiple international engineering characteristics. The index system takes the influencing factors of economic, social, environmental and technical benefits into account. In order to improve the rigidity and power of weight determination, we proposed the least squares method which combines the order relation method and the factor analysis method. Furthermore, the limitation of the one-way evaluation was effectively overcome by combining TOPSIS (Technique for Order Preference by Similarity to an Ideal Solution), grey relation analysis method and vector projection method. In addition, we adjusted the potential impact of the time on evaluation by using the quadratic weighted algorithm, so that we can dynamically evaluate the comprehensive benefits. Finally, we verified the established index system and evaluation model through an example of eight different investment plans of a transnational high voltage direct current (HVDC) transmission project. Altogether, results from this paper will provide a guidance reference and decision support for the grid corporation to invest in transnational power grid interconnection projects.

**Keywords:** transnational power grid interconnection project; comprehensive benefit evaluation; order relations method; factor analysis method; TOPSIS grey relation projection method; quadratic weighted algorithm based on temporal operator

---

## 1. Introduction

As the global economic community develops rapidly, energy consumption continues to grow. However, the extensive development and utilization of traditional fossil energy sources is a serious threat to human survival and development. The construction of transnational power grid interconnection projects is essential to accelerate the development and utilization of renewable energy in different countries and regions. It can provide strong support for external power supply in areas where electricity is not available. Therefore, regional economic integration, the optimal utilization of energy resources, and the improvement of power supply reliability can promote the interconnection of transnational power grids and make them sustainable.

The construction of such projects will bring huge economic, social, environmental and technical benefits. However, as a new research topic, the case of evaluating its comprehensive benefits is still

lacking. At present, the research concentrated on benefit evaluation of ordinary projects is relatively mature. Especially, taking economic benefit evaluation as an important aspect of project evaluation has been studied by many scholars. Wan J. et al. [1] pointed out that the economic benefits of the power grid are reflected in the three aspects of the increasing supply, reliability and loss of the power grid. Then, the calculation models are established separately to evaluate the power grid construction projects through the annual value of benefit cost. Pei Bie et al. [2] believed that the economic benefits brought by the wind power plant is modeled as an integrated savings run cost, and used the Probabilistic Load Flow-Based On Cumulant Method (PLF-CM) to calculate the savings run cost. More and more theoretical research and empirical analyses show that sustainable economic growth requires sustainable energy output as a support. Therefore, scholars turn their concerns to the social benefit evaluation of engineering projects. According to the specific characteristics of the power grid construction project, Xue M. et al. [3] established the social evaluation index system of the power grid construction project, which includes the social economy, the social environment and the natural environment. Liu Huifang et al. [4] established a comprehensive evaluation index system of social benefits of mineral resources development, through the analysis of national security degree, local social economy, social environment and social harmony. Song Q. et al. [5] checked from the internal mechanism of social welfare and related smart grid development, then used the Pressure-State-Response (PSR) model to evaluate the social welfare of the smart grid. Tian S. et al. [6] introduced the Life Cycle Cost (LCC) to analyze the cost of the ultra-high voltage grid. Moreover, she adopted the Shadow Price Theory (SPT) to evaluate the social benefits such as the optimization of resource allocation efficiency. With the increasing contradiction between energy supply and demand in the world, environmental pollution is becoming more and more serious. Therefore, environmental benefit has become a hot topic in the evaluation of engineering projects. Zhu F. et al. [7] believed that the regional power grid interconnection is a way to use natural resources for regional power generation, so that we can reduce greenhouse gas emissions. Zheng M. et al. [8] discussed the scheduling strategies and environmental benefits of residential electrical equipment in a microgrid. Feng F. et al. [9] evaluated Rice Straw-Based Synthetic Natural Gas (R-SNG). It was concluded that R-SNG had the environmental benefits of protecting non-biological resources and reducing greenhouse gas emission. With the adoption of new technology in engineering projects, more and more scholars pay attention to the technical benefits of project evaluation. Hernández J. C. et al. [10] evaluated the stability of frequency and voltage of the transmission system. Favuzza S. et al. [11] found that the interconnections of transmission systems in European and North African has brought new opportunities, such as "N-1" criterion was higher, electricity systems were safer and more reliable. Hernández J. C. et al. [12] studied two aspects that affect the stability of the power system, namely primary frequency control (PFC) and dynamic grid support (DGS).

With the advent of globalization and informatization, the evaluation of project benefit has been gradually developed from a single benefit index to comprehensive evaluation including economic, social, environmental and technical benefits [13]. However, most of these studies are static evaluations, that is to say, the comprehensive benefits of the project are evaluated by the mean value of different time periods. The disadvantage of these studies is that it is difficult to understand the dynamic changes and development trends of the comprehensive benefits of the projects. Singh A. et al. [14] proposed the economic and environmental evaluation of various power technologies and comprehensively considered the virtual operation of the existing power and product equipment. Paula et al. [15] discussed the use of reverse osmosis (RO) membranes, and concluded that using recycled membranes might bring economic and environmental benefits. The methods commonly used in the comprehensive evaluation are listed in Table 1. These evaluation methods have their own characteristics, but they often find it difficult to overcome their own limitations and still need further optimization.

**Table 1.** The methods commonly used in the comprehensive evaluation.

|  | Methods | Researcher |
| --- | --- | --- |
| Traditional evaluation | Triangular fuzzy function method | Zuo Y. et al. [16] |
|  | Analytic hierarchy process | Wu Q. et al. [17] |
|  | Matter-element extension | Liu Y. et al. [18] |
|  | Rumania Selection Method | Du Z. et al. [19] |
| Intelligent evaluation | Support Vector Machine | Zhang X. et al. [20] |
|  | Neural Networks | Ze-Hong L. I. [21] |
|  | Genetic Algorithm | Wen J. et al. [22] |
|  |  | Rongrong R. et al. [23] |
|  | Particle Swarm Algorithm | Zhao K. et al. [24] |
|  | Artificial Fish Swarm Algorithm | Zhou G. L. et al. [25] |

With the rapid development of China's electric power industry and the proposal of the 13th five-year plan for electric power development, the construction of a transnational power grid interconnection project is more scientific and reasonable. Regarding the aforementioned issues, this paper implements a comprehensive optimization and adjustment to the evaluation index system based on identifying the type of the object to be evaluated. In addition, this paper establishes an analytical method to identify the potential regions of such projects and proposes an integrative efficiency evaluation index system. In order to weaken the limitations of one-way evaluation and reflect the dynamic change of the comprehensive benefit, this paper combines TOPSIS (Technique for Order Preference by Similarity to an Ideal Solution), the grey relation analysis method and the vector projection method to calculate the grey relational projection proximity. Then, the dynamic comprehensive benefit development level was introduced, and the quadratic weighted calculation was carried out. In addition, this paper uses the least squares method combined order relation method and the factor analysis method to determine the weight, so as to improve the scientificity and rationality of the link to determine the weight of the index. On one hand, this paper can support grid companies worldwide to improve the investment decision-making level of such projects, improving investment strategy. On the other hand, it also advances the future development direction of energy in the interconnection planning, and effectively supports and serves the implementation of the Global Energy Interconnection strategy.

## 2. Methods

On the basis of comprehensive benefit evaluation of transnational grid interconnection projects, this paper puts forward a weighting method combining the order relations method and factor analysis method. In this paper, TOPSIS grey relation projection method is innovatively used to establish a comprehensive benefit evaluation model. The specific process is shown in Figure 1.

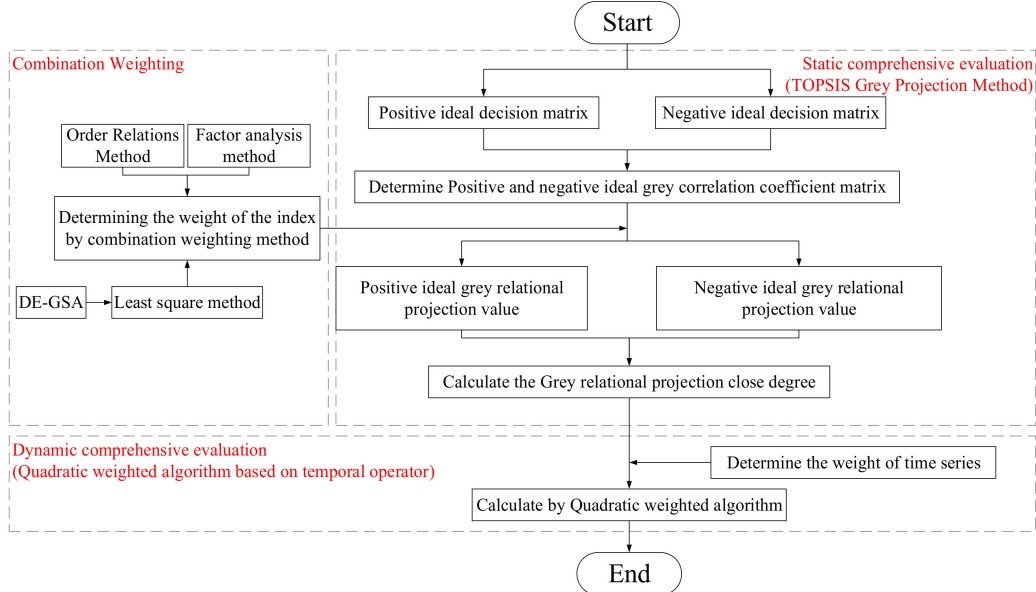

**Figure 1.** Flow chart of dynamic comprehensive benefit evaluation of such projects.

### *2.1. Combination Weighting Method*

#### 2.1.1. Order Relations Method

In the comprehensive benefit evaluation of transnational grid interconnection project, the emphasis is different according to the actual situation of the project, so the weight of each index can be changed flexibly with the change of decision-making problems. The order relation method can satisfy this requirement. The order relation method is a subjective weighting method without the consistency test, which avoids the overcomplicated calculation process, and can fully reflect the will of the experts [26]. The following steps constitute the order relations method:

Step 1: Determine Order Relations. The index order relation of the evaluation criterion is $Y_1 > Y_2 > \cdots Y_m$.

Step 2: Determine the relative importance degree of the adjacent index. The relative importance degree of the adjacent index can be described as:

$$R_j = Y_{j-1}/Y_j, \tag{1}$$

where $R_j$ is the ratio of the importance of expert evaluation indices $Y_{j-1}$ and $Y_j$ ($j = m, m-1, \cdots, 3, 2$).

Step 3: Determine the weight $u_j$, and the formula is obtained:

$$u_j = \left(1 + \sum_{j=2}^{m} \prod_{i=j}^{m} R_i\right)^{-1}, \tag{2}$$

$$u_{j-1} = R_j u_j \tag{3}$$

#### 2.1.2. Factor Analysis Method

Generally speaking, when the index system is constructed to evaluate the comprehensive benefits of transnational grid interconnection projects, the correlation between the indices may be strong, which may interfere with each other [27]. The factor analysis method can effectively solve the correlation between variables, and the whole operation process can be easily and quickly run through computer software such as SPSS [28]. The following steps constitute the factor analysis method:

Step 1: Converting the reverse index into the positive index, and then the z-score method is used to standardize the index.

$$Z_{ij} = \frac{z_{ij} - \overline{z_j}}{\sigma_j}, \tag{4}$$

where $\overline{z_j}$ is the mean of the index; $\sigma_j$ is the variance of the index.

Step 2: The extraction of factors. The cumulative variance contribution rate of the former $p$ common factor is shown below:

$$a_p = \sum_{i=1}^{p} \lambda_i / \sum_{i=1}^{m} \lambda_i, \tag{5}$$

where $\lambda_i$ is the characteristic root of the variable correlation coefficient matrix; $p$ is the number of common factors, which can be determined according to the basic principle of $a_p \geq 85\%$ and $\sum_{i=1}^{p} \lambda_i > 1$.

Step 3: The name and the rotation of the factor. By changing the axis, each factor explains the ratio of the variance of the original variable that is redistributed, which makes it easier for people to understand the factors that have been extracted.

Step 4: Determining the weight. On the basis of the factor score and comprehensive evaluation value, the factor score of each index is calculated as follows:

$$Y = v_1 z_1 + v_2 z_2 + \cdots v_m z_m, \tag{6}$$

where $V = (v_1, v_2, \cdots, v_m)$ is the weight which determined by the factor analysis method.

### 2.1.3. Least Squares Method

Since the subjective weighting method embodies the value of the index and the objective weighting method reflects the information of the index, each has its own characteristics, so the comprehensive evaluation should reflect the unity of the two. The combined weights of each index are $W = [w_1, w_2, \cdots w_m]^T$.

Since the smaller the evaluation value of all the indices under the subjective weighting, the better. Therefore, the combinational evaluation model optimized by the least square method is established. The model is constructed as:

$$
\begin{aligned}
\min H(w) &= \sum_{i=1}^{n} \sum_{j=1}^{m} \left\{ \left[ (u_j - w_j) z_{ij} \right]^2 + \left[ (v_j - w_j) z_{ij} \right]^2 \right\} \\
s.t. \sum_{j=1}^{m} w_j &= 1, \ w_j \geq 0 (j = 1, 2, \cdots, m)
\end{aligned} \tag{7}
$$

where $m$ is the number of the indices; $n$ is the number of objects.

### 2.2. TOPSIS Grey Projection Method

The basic principle of the TOPSIS method is to find out the positive and negative ideal scheme respectively in the normalized decision matrix. Then, based on them we can calculate the approximation degree between the feasible solution and the two benchmark solutions, so as to evaluate the advantages and disadvantages of each feasible solution [29].

After the original data was positively normalized and non-dimension treated, at the time $t$, the maximum value of each evaluated object in the index $j$ was considered as the positive ideal solution $Z^+(t) = \{z_{01}^+(t), \cdots, z_{0n}^+(t)\}$, and the minimum of which was considered as the negative ideal solution $Z^-(t) = \{z_{01}^-(t), \cdots, z_{0n}^-(t)\}$.

Step 1: Determine the positive and negative ideal grey correlation coefficient matrix.

$$\delta_{ij}(t) = \frac{\min\limits_{i}\min\limits_{j}\left|z_{0j}^*(t) - z_{ij}(t)\right| + \rho\max\limits_{i}\max\limits_{j}\left|z_{0j}^*(t) - z_{ij}(t)\right|}{\left|z_{0j}^*(t) - z_{ij}(t)\right| + \rho\max\limits_{i}\max\limits_{j}\left|z_{0j}^*(t) - z_{ij}(t)\right|}. \tag{8}$$

Through the formula above, it can calculate the positive ideal grey correlation coefficient matrix $E^+(t) = \left\{\delta_{ij}^+(t)\right\}$ and the negative one $E^-(t) = \left\{\delta_{ij}^-(t)\right\}$.

Step 2: Assume that the angle between the feasible solution $a_i(t) = (\delta_{i1}(t)w_1(t), \cdots, \delta_{im}(t)w_m(t))$ and ideal solution $a_i^*(t) = (w_1(t), \cdots, w_m(t))$ is $\theta_i(t)$, and the cosine is calculated. Then, the value of the positive ideal grey relational projection is $D_i^+(t) = \sum\limits_{j=1}^{n}\delta_{ij}^+(t)\overline{w_j(t)}$. Similarly, the value of the negative ideal grey relational projection can be obtained. Where $\overline{w_j(t)} = w_j^2(t) / \sqrt{\sum\limits_{j=1}^{m}w_j^2(t)}$.

Furthermore, the correlation of grey relational projection is $y_i(t) = \frac{D_i^{+2}(t)}{D_i^{+2}(t) + D_i^{-2}(t)}$. It can be proved that the larger $y_i(t)$ is, the closer the feasible solution is to the ideal solution [30].

### 2.3. Quadratic Weighted Algorithm Based on Temporal Operator

The quadratic weighted calculation algorithm is based on the first weighted calculation, which highlights the role of time and uses the time-serial mean operators. The specific steps are as follows:

Step 1: The value of object $i$ at time $t$, $y_i(t)$ can be obtained from the first weighted static evaluation.

Step 2: Determine the weight of the time series.

$$\max\left(-\sum\limits_{k=1}^{t} r_k \cdot \ln r_k\right)$$
$$s.t. \begin{cases} \sum\limits_{k=1}^{t}\frac{t-k}{t-1}r_k = \lambda, \\ \sum\limits_{k=1}^{t} r_k = 1 \end{cases} \tag{9}$$

where $R = (r_1, r_2, \cdots r_t)^T$ is the time weight vector which can be obtained by solving the nonlinear programming problem above; $\lambda$ represents the importance of time, $\lambda \in (0,1)$. The closer the value of $\lambda$ is to 0, the greater importance the evaluator attaches to the later stage data; the closer the value of $\lambda$ is to 1, the greater importance the evaluator attaches to the earlier stage data [31].

## 3. Comprehensive Benefit Evaluation Index System of Transnational Power Grid Interconnection Project

### 3.1. Comprehensive Benefit Evaluation Index System

As an international large-scale construction project, the transnational power grid interconnection project is often closely related to the energy supply, infrastructure and national defense and people's livelihood of the target country or region. The main principles of building the comprehensive benefit evaluation index system of such projects are described as follow:

(1) This paper is based on the characteristics of high complexity and various factors of transnational interconnection project.

(2) Screening index should be based on the principles of combining qualitative with quantitative analysis.

(3) We should adhere to the scientific nature, rationality, practicality and operability of comprehensive evaluation index. In addition, the relevant technical standards and industry-wide standards in the fields of related power technology, economic, social and environment at home and abroad are taken as the reference basis.

The comprehensive benefit evaluation index system is listed in Table 2.

**Table 2.** Comprehensive Benefit Evaluation Index System of such projects.

| Object | First-Level Index | Second-Level Index |
| --- | --- | --- |
| Comprehensive Benefit of the Transnational Power Grid Interconnection Project G | Economic Benefit A | Return on investment A1 |
| | | Power Price Competition A2 |
| | | Forex expansion ratio in export A3 |
| | Social Benefit B | Employment effect B1 |
| | | Direct contribution rate of GDP B2 |
| | | International friendship degree B3 |
| | Environmental Benefit C | Comprehensive energy consumption efficiency C1 |
| | | Proportion of clean energy C2 |
| | | Carbon dioxide emission reduction C3 |
| | Technical Benefit D | Cable adoption rate D1 |
| | | "N-1" criterion D2 |
| | | Reliability rate of power supply D3 |

### 3.2. Analysis of the Benefit Evaluation Index

#### 3.2.1. Economic Benefit

Economic benefit evaluation is on the basis of the financial analysis, and then using the comprehensive evaluation method to draw a general conclusion about the process and effects of financial activities. In this paper, the economic benefit can mainly be analyzed from the perspectives of return on investment, Power Price Competition, Forex expansion ratio in export, and debt service coverage ratio.

(1) Return on Investment

The return on investment (ROI) includes annual profit and total investment, which refers to the value that should be returned through investment. It can also be understood as the economic return of an enterprise from an investment activity. In addition, because of eliminating the profit difference caused by the different amount of investment, it is conducive to judge the performance of the investment center. Moreover, ROI can be used as the basis for choosing investment opportunities, and is helpful to optimize the allocation of resources.

$$ROI = \frac{AP}{I_D} \times 100\%, \tag{10}$$

where $I_D$ is the total investment; $AP$ is the annual profit.

(2) Power Price Competition

The transnational power price competition tariff measures the price advantage of imported power products over similar types of power products in power input countries. The specific calculation method is as follows:

$$CS_t = T_r - T_s, \tag{11}$$

where $CS_t$ is the power price competition; $T_r$ is the power price at the receiver end; $T_s$ is the sending-oriented power price. If $T_s$ is lower than $T_r$, it means that the construction of the project has the competitiveness of power price and can effectively curb the rise of power price.

(3) Forex Expansion Ratio in Export

The forex expansion ratio in export is also known as the appreciation rate of foreign exchange, which can be defined as the ratio of $Ni_e$ and $Ec_i$, thus assessing the economic benefits of the processing with imported materials.

$$FER = \frac{Ni_e - Ec_i}{Ec_i} \times 100\%,\tag{12}$$

where $Ni_e$ is the net income of foreign exchange for export products; $Ec_i$ is the foreign exchange cost of imported raw materials.

### 3.2.2. Social Benefit

Social benefit comprises the impact on society and contribution to society. This paper evaluates the social benefits of the distribution network from the aspects of employment effect, the direct contribution rate of GDP, reliability rate of the power supply, new technology diffusion benefit and international friendship degree.

(1) Employment Effect

The construction and operation of the power grid project involve the design, construction, maintenance and other industries, which provide direct or indirect employment opportunities for the society.

$$E_e = N_e / I_D,\tag{13}$$

where $E_e$ is the employment effect; $N_e$ is the amount of new employment in this project; $I_D$ is the direct investment.

(2) Direct Contribution Rate of GDP

Developing economy is a basic state policy of our country. The contribution to the economic growth of a region is one of the critical criteria to inspect the advantages and disadvantages of the project. Regional economic growth is the result of the combined effects of various factors, such as policy adjustment, technological progress and investment increase. However, the construction and the operation of power grid is the basic conditions to promote the economic growth of a region. The direct contribution rate of GDP can determine the influence of transnational power grid interconnection projects on the national economy.

$$S = \frac{C}{G} \times 100\%,\tag{14}$$

where $S$ is the direct contribution rate of GDP; $C$ is the annual revenue of electricity sales; $G$ is the GDP of the corresponding year.

(3) International Friendship Degree

International friendship degree refers to the friendly relations between countries connected by power grids. This index can be used to obtain the beneficiary's understanding and preference of the power grid interconnection countries by questionnaire survey, and estimate the number of plans to study, work or travel to such countries in the next three years. Finally, the improvement value of international friendly relations brought by the project is calculated.

### 3.2.3. Environmental Benefit

Environmental benefits refer to the impact on the human living environment. Combined with the characteristics of transnational power grid interconnection project, this paper evaluates it from the aspects of comprehensive energy consumption efficiency, the proportion of clean energy and carbon dioxide emission reduction.

(1) Comprehensive Energy Consumption Efficiency

The comprehensive energy consumption efficiency reflects the energy-saving effect of a project construction. The implementation of the transnational power grid interconnection project can directly or indirectly enhance the comprehensive energy consumption efficiency. If the comprehensive energy consumption efficiency of the project is lower than the average energy consumption level of the society, then the project has a better energy-saving effect.

$$E' = E/O, \tag{15}$$

where $E\prime$ is the energy consumption efficiency of the year; $E$ is the value of comprehensive energy consumption in the year of the project; $O$ is the net output of the company in the current year.

(2) Proportion of Clean Energy

Promoting the "double alternative" strategy and forming an energy pattern with clean energy as the leading factor determine the key role of power technology in future energy development. Its core is to continuously improve the efficiency and economics of clean energy development, including wind-generated electricity, solar power generation and Distributed Generation (DG) technology. The proportion of clean energy should be considered in the transnational power grid interconnection project, which can be used in the calculation of the proportion of the electricity generated by clean energy to the total transmission electricity in the process of transmission. Therefore, the contribution of such projects to energy conservation and emission reduction will be calculated.

(3) Carbon Dioxide Emission Reduction

The establishment of the transnational power grid interconnection projects is conducive to the development and utilization of renewable energy, and can reduce a large amount of pollutants and greenhouse gas emissions [32]. In addition, the United Nations Framework Convention on Climate Change (UNFCCC) will be guaranteed to control the global average temperature rise within 2 °C in 2050.

$$\theta^* = \theta' - \theta, \tag{16}$$

where $\theta^*$ is the carbon dioxide emission reduction; $\theta'$ is the total amount of carbon dioxide emissions reduction after project construction; $\theta$ is the carbon dioxide emissions reduced by the average power supply of the power industry in the receiving end state.

3.2.4. Technical Benefit

(1) Cable Adoption Rate

A fully-interconnected transnational power grid project with more cross-border interconnections, greater storage potential is the goal of China's power grid. To achieve this, we need to rely on cables to complete the transmission of electricity. The cable adoption rate can represent the level of equipment which affects the reliability of the power supply.

(2) "N-1" Criterion

In the case of transmission interruption and variable renewable supply, network operators have to maintain a reliability margin to avoid interruption and blackouts of transnational power grid interconnection projects. Transmission line is the basic equipment of a power grid, and its blackout will cause subsequent cascading outages. The reliability margin is presently determined by the N-1 reliability criterion.

(3) Reliability Rate of Power Supply

The realistic purpose of developing transnational power grid interconnection projects is to construct efficient, safe and stable transmission across countries or regions, and to realize the

interconnection and interoperability of power grids between different countries or regions. After the interconnection of power grids, the reliability rate of power supplies is improved and the loss of outage is reduced because the reserve capacity of each power system can support each other. The formula for calculating the reliability of power supply is as follows:

$$RSI = \left(1 - \frac{t}{T}\right) \times 100\%, \tag{17}$$

where $RSI$ is the reliability rate of power supply; $t$ is the average interruption hours of customers; $T$ is the time period of the statistic.

## 4. Case Study

### 4.1. Case Background and Data Sources

On the basis of the existing relevant case background in the literature [33], this paper improves and supplements the necessary parameters and data which are not clearly given in the data. Then, the evaluation index system and evaluation model established in this paper are used to evaluate the comprehensive benefits of eight plans of transnational power grid interconnection projects. These eight plans are implemented by eight power companies, which use different voltage levels, wire thickness and construction investment to connect different power plants at the receiving terminal. The operation period of each plan is 10 years. Finally, one of the plans needs to be chosen for implementation, so as to make a brief analysis of the evaluation results of the eight plans.

### 4.2. Static Comprehensive Evaluation Results

In this section, the comprehensive weight (Table 3) of each year is determined by the combination weighting method which combines the order relations method and factor analysis method. For the solution of the least squares method, this paper put forward a global optimization method based on the Differential Evolution Algorithm (DE) and Gravitational Search Algorithm (GSA). By introducing the mutation strategy of group search and synergistic search ability, the algorithm not only improves search ability of GSA in the process of gravitational constant optimization, but also keeps the diversity of the population. Therefore, GSA is prevented from falling into a local solution [34].

**Table 3.** Comprehensive weight $w_j$ based on the combination weighting method.

| Index | $t_1$ | $t_2$ | $t_3$ | $t_4$ | $t_5$ | $t_6$ | $t_7$ | $t_8$ | $t_9$ | $t_{10}$ |
|-------|-------|-------|-------|-------|-------|-------|-------|-------|-------|----------|
| A1 | 0.074 | 0.070 | 0.050 | 0.117 | 0.074 | 0.060 | 0.109 | 0.056 | 0.099 | 0.063 |
| A2 | 0.052 | 0.078 | 0.078 | 0.050 | 0.082 | 0.050 | 0.092 | 0.055 | 0.076 | 0.096 |
| A3 | 0.05 | 0.051 | 0.089 | 0.127 | 0.050 | 0.067 | 0.084 | 0.075 | 0.050 | 0.056 |
| B1 | 0.064 | 0.050 | 0.067 | 0.063 | 0.100 | 0.050 | 0.106 | 0.085 | 0.060 | 0.107 |
| B2 | 0.115 | 0.062 | 0.149 | 0.061 | 0.050 | 0.050 | 0.050 | 0.061 | 0.097 | 0.057 |
| B3 | 0.105 | 0.058 | 0.050 | 0.050 | 0.064 | 0.103 | 0.058 | 0.050 | 0.135 | 0.069 |
| C1 | 0.058 | 0.050 | 0.105 | 0.065 | 0.148 | 0.064 | 0.053 | 0.061 | 0.057 | 0.050 |
| C2 | 0.091 | 0.086 | 0.111 | 0.055 | 0.063 | 0.088 | 0.070 | 0.142 | 0.137 | 0.089 |
| C3 | 0.102 | 0.111 | 0.109 | 0.117 | 0.078 | 0.105 | 0.111 | 0.058 | 0.077 | 0.083 |
| D1 | 0.091 | 0.136 | 0.057 | 0.065 | 0.092 | 0.176 | 0.153 | 0.122 | 0.062 | 0.159 |
| D2 | 0.103 | 0.139 | 0.084 | 0.157 | 0.149 | 0.117 | 0.050 | 0.062 | 0.074 | 0.102 |
| D3 | 0.096 | 0.110 | 0.050 | 0.073 | 0.050 | 0.072 | 0.068 | 0.173 | 0.076 | 0.069 |

Take the first year of operation period as an example, this paper calculates the positive ideal grey correlation coefficient matrix $E^+(t)$ and the negative one $E^-(t)$ when $t = 1$ by Formula (10). The results are shown in Tables 4 and 5.

**Table 4.** The positive ideal grey correlation coefficient matrix when $t = 1$.

| Index | P1 | P2 | P3 | P4 | P5 | P6 | P7 | P8 |
|-------|-----|-----|-----|-----|-----|-----|-----|-----|
| A1 | 0.673 | 0.922 | 0.604 | 0.757 | 1.000 | 0.733 | 0.639 | 0.817 |
| A2 | 0.783 | 0.690 | 0.743 | 0.906 | 0.743 | 0.906 | 1.000 | 0.673 |
| A3 | 0.765 | 0.692 | 0.838 | 0.791 | 0.753 | 1.000 | 0.927 | 0.745 |
| B1 | 0.575 | 0.634 | 0.591 | 0.695 | 0.673 | 0.769 | 0.743 | 1.000 |
| B2 | 0.582 | 0.724 | 1.000 | 0.876 | 0.651 | 0.629 | 0.711 | 0.896 |
| B3 | 0.478 | 0.592 | 0.521 | 1.000 | 0.613 | 0.442 | 0.506 | 0.744 |
| C1 | 0.998 | 0.999 | 1.000 | 1.000 | 0.998 | 1.000 | 1.000 | 1.000 |
| C2 | 0.428 | 0.450 | 0.649 | 0.516 | 1.000 | 0.762 | 0.444 | 0.428 |
| C3 | 0.333 | 1.000 | 0.394 | 0.498 | 0.429 | 0.525 | 0.362 | 0.585 |
| D1 | 0.959 | 0.856 | 0.814 | 0.815 | 0.868 | 1.000 | 0.987 | 0.854 |
| D2 | 0.764 | 0.856 | 0.717 | 0.743 | 0.981 | 0.695 | 0.828 | 1.000 |
| D3 | 0.871 | 1.000 | 0.797 | 0.808 | 0.881 | 0.839 | 0.800 | 0.953 |

**Table 5.** The negative ideal grey correlation coefficient matrix when $t = 1$.

| Index | P1 | P2 | P3 | P4 | P5 | P6 | P7 | P8 |
|-------|-----|-----|-----|-----|-----|-----|-----|-----|
| A1 | 0.855 | 0.637 | 1.000 | 0.750 | 0.604 | 0.774 | 0.917 | 0.699 |
| A2 | 0.828 | 0.967 | 0.878 | 0.724 | 0.878 | 0.724 | 0.673 | 1.000 |
| A3 | 0.879 | 1.000 | 0.799 | 0.848 | 0.895 | 0.692 | 0.732 | 0.907 |
| B1 | 1.000 | 0.861 | 0.956 | 0.769 | 0.797 | 0.695 | 0.718 | 0.575 |
| B2 | 1.000 | 0.748 | 0.582 | 0.634 | 0.845 | 0.886 | 0.763 | 0.624 |
| B3 | 0.853 | 0.635 | 0.744 | 0.442 | 0.613 | 1.000 | 0.777 | 0.521 |
| C1 | 1.000 | 1.000 | 0.998 | 0.999 | 1.000 | 0.999 | 0.998 | 0.998 |
| C2 | 1.000 | 0.898 | 0.557 | 0.715 | 0.428 | 0.494 | 0.921 | 1.000 |
| C3 | 1.000 | 0.333 | 0.685 | 0.502 | 0.599 | 0.477 | 0.810 | 0.437 |
| D1 | 0.843 | 0.943 | 1.000 | 0.999 | 0.929 | 0.814 | 0.823 | 0.945 |
| D2 | 0.885 | 0.787 | 0.958 | 0.915 | 0.704 | 1.000 | 0.813 | 0.695 |
| D3 | 0.903 | 0.797 | 1.000 | 0.983 | 0.893 | 0.940 | 0.996 | 0.829 |

Finally, this paper calculates the close degree of grey relational projection $y_i(t)$ (Table 6) of the eight engineering schemes within 10 years of the operation period in the case background.

**Table 6.** Close degree of the grey relational projection.

| Index | $t_1$ | $t_2$ | $t_3$ | $t_4$ | $t_5$ | $t_6$ | $t_7$ | $t_8$ | $t_9$ | $t_{10}$ |
|-------|-------|-------|-------|-------|-------|-------|-------|-------|-------|----------|
| P1 | 0.410 | 0.491 | 0.503 | 0.464 | 0.484 | 0.475 | 0.448 | 0.451 | 0.545 | 0.511 |
| P2 | 0.512 | 0.497 | 0.508 | 0.477 | 0.505 | 0.463 | 0.527 | 0.464 | 0.484 | 0.529 |
| P3 | 0.467 | 0.510 | 0.462 | 0.475 | 0.540 | 0.547 | 0.510 | 0.554 | 0.510 | 0.475 |
| P4 | 0.510 | 0.473 | 0.469 | 0.495 | 0.479 | 0.515 | 0.486 | 0.497 | 0.567 | 0.523 |
| P5 | 0.514 | 0.492 | 0.511 | 0.527 | 0.499 | 0.527 | 0.567 | 0.496 | 0.546 | 0.514 |
| P6 | 0.470 | 0.500 | 0.534 | 0.528 | 0.486 | 0.525 | 0.522 | 0.475 | 0.433 | 0.532 |
| P7 | 0.454 | 0.485 | 0.533 | 0.488 | 0.488 | 0.477 | 0.530 | 0.488 | 0.407 | 0.445 |
| P8 | 0.528 | 0.547 | 0.465 | 0.461 | 0.492 | 0.552 | 0.482 | 0.527 | 0.474 | 0.492 |

*4.3. Dynamic Comprehensive Evaluation Results*

In order to highlight the role of time, $\lambda = 0.1$ is obtained by Formula (11), which means that we attach great importance to the later stage data when evaluating the project. The weights of time series are obtained as follows:

$$R = (0.00065, 0.00137, 0.00288, 0.00605, 0.01274, 0.02680, 0.05636, 0.11861, 0.24954, 0.52410)$$

The time-serial mean operator arithmetic was used to do the quadratic weighted calculation of the value of the static evaluation. The value of dynamic comprehensive evaluation can be calculated by Formula (18).

$$h_i = \sum_{k=1}^{t} y_i(t) \cdot r_k \tag{18}$$

In this way, the value of the dynamic comprehensive evaluation of eight projects over 10 years (Figure 2) is obtained.

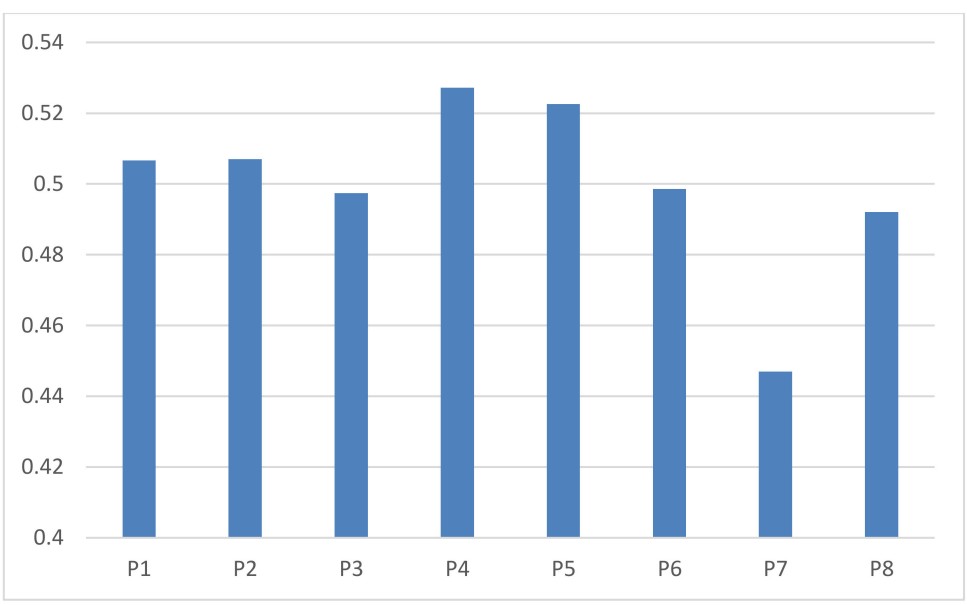

**Figure 2.** Dynamic comprehensive evaluation results.

*4.4. Result Analysis*

According to the dynamic evaluation results of Figure 2, the project plan *P*4 was selected and the result was 0.5272. In addition, with the help of Figure 3, the dynamic trend of comprehensive benefits of eight projects over 10 years was analyzed in detail.

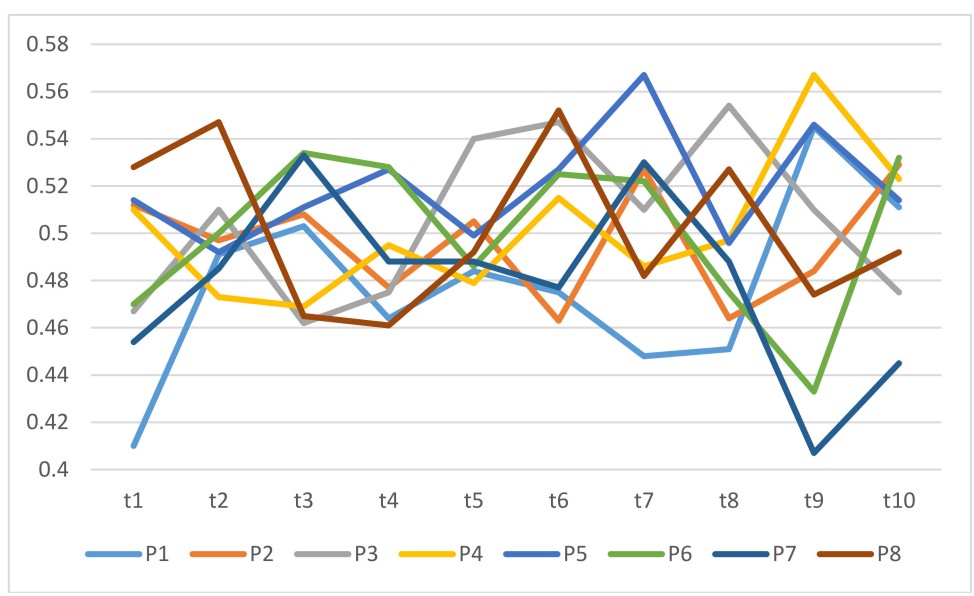

**Figure 3.** The dynamic trend of comprehensive benefits of eight projects over 10 years.

From the static evaluation results of the cross section of eight projects in a certain year, we can conclude that the corresponding annual optimal project will be different due to the different emphasis of each project and the different control of future trends. For example, the plan $P8$ in the time $t_1$, $t_2$, $t_6$ is optimal during the operation period; the plan $P6$ in the time $t_3$, $t_4$, $t_{10}$ is optimal; the plan $P3$ in the time $t_5$, $t_8$ is optimal; the plan $P5$ in the time $t_7$ is optimal; the plan $P4$ in the time $t_9$ is optimal and the result is 0.567. In addition, the optimal plan is different each year, so it is impossible to determine the exact solution accurately. Assuming that we attach great importance to the data of the later three years when evaluating the project, according to the static evaluation results, the optimal plan will be simplified into three plans, namely, $P3$, $P4$, $P6$. Then, according to the dynamic evaluation results of a project over 10 years, the comprehensive benefit of the plan $P4$ is on the rise, which accords with the requirement of paying great attention to the later benefit of the project in the process of dynamic comprehensive evaluation. In addition, the comprehensive benefit of the plan $P5$ is relatively high in each year, ranking it second among the eight plans, with a result of 0.5226. To sum up, these results further illustrate the applicability and scientificity of the dynamic comprehensive benefit evaluation method proposed in this paper.

## 5. Conclusions

On 14 May 2017, Chinese Electricity Council Chairman Liu in the "One Belt and One Road" international cooperation summit, BBS pointed out that the global clean energy reserves are very abundant, and only 5/10,000 can meet human energy needs. The construction of the global energy Internet has huge comprehensive benefits. It can realize the economic growth and promote the development of the smart grid, UHV, new energy and other emerging industries. Firstly, based on the globalization perspective, this paper sets up an index system for evaluating the comprehensive benefits of transnational power grid interconnection projects, including 12 indices. Secondly, it constructs a dynamic comprehensive evaluation model based on the subjective and objective combinatorial portfolio weighting method and TOPSIS Grey Projection Method, so that the evaluation model is more applicable and scientific. Finally, in the current stage of the global energy interconnection, this paper can provide some guidance for the government power sector, construction units, investment enterprise and other different participants in the relevant areas of business activities or decision-making.

**Author Contributions:** Methodology: X.T.; Writing original draft: J.X.; Writing review & editing: J.L.

**Funding:** This research was funded by the science and technology project of National Electric Net Ltd. [Research on Comprehensive evaluation Method of economic, social and environmental benefits of the Transnational Power Grid Interconnection project] grant number [SGTYHT/16-JS-198].

**Conflicts of Interest:** The authors declare no conflict of interest.

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
