# Peer review of "Dynamic Comprehensive Benefit Evaluation of the Transnational Power Grid Interconnection Project Based on Combination Weighting and TOPSIS Grey Projection Method"

_sustainability, doi:10.3390/su10124672_

Round 1

Reviewer 1 Report

The paper presents the benefit evaluation of the transnational power grid interconnection projects. The research is timely, adds important insight to the current discussion about this topic.  However, the present manuscript lacks of the inclusion of important parameters that impact on these benefits. Thus, economic, social and environmental benefits are only analysed. However, technical benefits are not accounted; see ref. (New Energy Corridors in the Euro-Mediterranean Area: The Pivotal Role of Sicily, Energies 11(6) 2018)

For example this kind of projects can improve the local stability of the power system in national areas when a large amount of renewable energies (e.g. PV, and wind power, electric vehicles) are involved. Some of these technical benefits should be included in the model. According to these issues, the authors are invited to update the introduction and refer the following reference in the literature review: (i) Enhanced utility-scale photovoltaic units with frequency support functions and dynamic grid support for transmission systems, IET Renewable Power Generation 11 (3), 361-372, 2017; (ii) Primary frequency control and dynamic grid support for vehicle-to-grid in transmission systems

International Journal of Electrical Power & Energy Systems 100, 152-166, 2018

Author Response

Thank you very much for your letter and advice on our manuscript. We have resubmitted new version of paper in accordance with recommendations of the reviewers. We have addressed the comments raised by the reviewers, and the amendments are highlighted in red in the revised manuscript. We have made a serious revision to every question raised by the reviewers, which is as follows:

Point: The paper presents the benefit evaluation of the transnational power grid interconnection projects. The research is timely, adds important insight to the current discussion about this topic.  However, the present manuscript lacks of the inclusion of important parameters that impact on these benefits. Thus, economic, social and environmental benefits are only analysed. However, technical benefits are not accounted; see ref. (New Energy Corridors in the Euro-Mediterranean Area: The Pivotal Role of Sicily, Energies 11(6) 2018)

For example this kind of projects can improve the local stability of the power system in national areas when a large amount of renewable energies (e.g. PV, and wind power, electric vehicles) are involved. Some of these technical benefits should be included in the model. According to these issues, the authors are invited to update the introduction and refer the following reference in the literature review: (i) Enhanced utility-scale photovoltaic units with frequency support functions and dynamic grid support for transmission systems, IET Renewable Power Generation 11 (3), 361-372, 2017; (ii) Primary frequency control and dynamic grid support for vehicle-to-grid in transmission systems. International Journal of Electrical Power & Energy Systems 100, 152-166, 2018

Response: We refer to three paper you provide us, and we updated the introduction and referred the following reference in the literature review: (1) Favuzza, S., Ippolito, M., Massaro, F., Mineo, L., Musca, R., & Zizzo, G. (2018). New energy corridors in the euro-mediterranean area: the pivotal role of sicily. Energies, 11(6), 1415. (2) Hernández, J. C., Bueno, P. G., & Sanchez-Sutil, F. (2017). Enhanced utility-scale photovoltaic units with frequency support functions and dynamic grid support for transmission systems. Iet Renewable Power Generation, 11(3), 361-372. (3) Hernández, J. C., Sanchez-Sutil, F., Vidal, P. G., & Rus-Casas, C. (2018). Primary frequency control and dynamic grid support for vehicle-to-grid in transmission systems. International Journal of Electrical Power & Energy Systems, 100, 152-166. At the same time, we added the technical benefit index into the comprehensive benefit index system of the transnational power grid interconnection project. On the basis of economic, social and environmental benefits, we fully considered the technical benefit brought by the transnational power grid interconnection project, such as Cable adoption rate, N-1” criterion and Reliability rate of power supply.

Finally, we went through a complete English check and typographical carefully to avoid typos or grammatical errors. We hope that the revision is acceptable and look forward to hearing from you soon.

With best wishes,

Jiaming Xu

17551595449@163.com

Reviewer 2 Report

The paper presents an evaluation index system based on the least squares method, combined order relation method and the factor analysis method.  A gravitational search algorithm optimized by differential evolution (DE-GSA) is used for optimization.

The topic of paper is interesting. However, the paper lacks clarity and presentation which makes it very hard to follow.

The language of the paper is very hard to follow. There are numerous grammatical errors and many ambiguous sentences. Some comments:

1.      Abstract lacks clarity and contributions of the paper.

2.      In introduction, the literature gap is not identified properly and the contributions of the manuscript are not presented clearly.

3.      Organisation of paper is hard to follow.

In the present form the paper is not suitable for publication.

Author Response

Thank you very much for your letter and advice on our manuscript. We have resubmitted new version of paper in accordance with recommendations of the reviewers. We have addressed the comments raised by the reviewers, and the amendments are highlighted in red in the revised manuscript. We have made a serious revision to every question raised by the reviewers, which is as follows:

Point 1: Abstract lacks clarity and contributions of the paper.

Response 1: We have revised the abstract to highlight the contributions of this paper, including new objects to be evaluated and improved evaluation models.

Point 2: In introduction, the literature gap is not identified properly and the contributions of the manuscript are not presented clearly.

Response 2: In introduction, we revised the introduction, including identified properly the literature gap, added shortcomings in the literature, and presented clearly the contributions of the manuscript.

Point 3: Organisation of paper is hard to follow.

Response 3: We revised the abstract and introduction, presented clearly the background and contributions of the manuscript, and made organization of paper is easy to follow.

Finally, we went through a complete English check and typographical carefully to avoid typos or grammatical errors. We hope that the revision is acceptable and look forward to hearing from you soon.

With best wishes,

Jiaming Xu

17551595449@163.com

Round 2

Reviewer 1 Report

This revision has addressed all raised issues in the original version and can be acceptable for publication.

Author Response

Thank you very much for your letter and advice on our manuscript.

Point: This revision has addressed all raised issues in the original version and can be acceptable for publication.

Response: We would like to thank you for the positive and constructive comments and suggestions.

We look forward to hearing from you soon.

With best wishes,

Jiaming Xu

17551595449@163.com

Reviewer 2 Report

Thanks for the revisions. Authors have responded satisfactorily to most of my comments.

However, still the langauge of manuscript is not clear. There are many long and ambigious sentences throughout the manuscript (even in abstract) which make it very hard to follow.

For eg. "At the same time, not only in order to improve the scientificity and rationality of the link to determine the weight, this paper uses the least squares method combined order relation method and the factor analysis method to determine the weight, but also for the solution of the least squares model, we proposes a gravitational search algorithm optimized by differential evolution (DE-GSA) to effectively prevent the optimization algorithm from falling into local solutions. "

Authors are advised to improve the langauge of the paper, before it can be accepted for possible publication.

Author Response

Thank you very much for your letter and advice on our manuscript. We have resubmitted new version of paper in accordance with recommendations of the reviewers. We have addressed the comments raised by the reviewers, and the amendments are highlighted in red in the revised manuscript. We have made a serious revision to every question raised by the reviewers, which is as follows:

Point: Thanks for the revisions. Authors have responded satisfactorily to most of my comments.

However, still the langauge of manuscript is not clear. There are many long and ambigious sentences throughout the manuscript (even in abstract) which make it very hard to follow.

For eg. "At the same time, not only in order to improve the scientificity and rationality of the link to determine the weight, this paper uses the least squares method combined order relation method and the factor analysis method to determine the weight, but also for the solution of the least squares model, we proposes a gravitational search algorithm optimized by differential evolution (DE-GSA) to effectively prevent the optimization algorithm from falling into local solutions." Authors are advised to improve the language of the paper, before it can be accepted for possible publication.

Response: We went through a complete English check and typographical carefully to avoid typos or grammatical errors.

We hope that the revision is acceptable and look forward to hearing from you soon.

With best wishes,

Jiaming Xu

17551595449@163.com
